# POLYNOMIAL-TIME REASONING AT THE EDGE OF NP

## ABSTRACT

Large language models (LLMs) are fundamentally P-time machines, yet they show a surprising ability to solve small instances of NP-hard problems. This capability, however, collapses as complexity grows, posing a fundamental challenge: can a P-time machine effectively engage with problems presumed to be outside of P? Our approach confronts this challenge by reframing the LLM's role. Instead of a monolithic reasoner tasked with generating a complete solution in one go, we leverage it as a P-time heuristic function. In this framework, inference is scaled by increasing the number of search calls, with each call deploying the LLM as a P-time machine to make a local, heuristic decision. We implement this paradigm using scalable algorithms like Reflective Search and MCTS, which systematically explore the solution space. Our evaluation on several NP tasks demonstrates that this LLM-guided search approach maintains robust polynomial scalability, delivering strong approximate solutions where a direct approach would fail. Further analysis reveals key properties of this framework, such as its significant performance gains from high-quality initial solutions, which underscores the synergy between heuristic guidance and structured exploration. These findings establish that the viable path for P-time computation to navigate intractable NP landscapes lies in the structured integration of LLMs as heuristic guides within classical, scalable search algorithms.

## 1 INTRODUCTION

The relationship between P and NP is a central question in computer science. Large language models (LLMs) provide a new empirical lens on this question. At their core, LLMs are P-time machines: they are built from polynomial-time tensor operations and process polynomial-length contexts, consistent with the *circuit hypothesis* (Elhage et al., 2021; Nanda et al., 2023; Shi et al., 2024) which views a transformer as a polynomial circuit for a given task. Scaling LLM computation at test-time—by generating longer reasoning chains or performing more inference steps—is analogous to expanding this polynomial circuit (Li et al., 2024; Merrill & Sabharwal, 2025).

Despite their P-time nature, LLMs demonstrate a strong ability to solve small instances of NP problems like SAT through simple prompting (Hazra et al., 2024). However, this direct solving approach fails to scale (Fan et al., 2024a). As problem complexity increases, the performance of LLMs collapses. This is expected: the fixed computational complexity of a single inference is inherently insufficient for navigating an exponential search space. To effectively tackle these problems, the computation must be scaled in a principled manner. We propose a structured search paradigm as a principled way to achieve this scaling. Instead of treating the model as a monolithic solver attempting to find a "needle in an exponential haystack in a single pass," we deploy the LLM as a P-time heuristic component. Our framework achieves scaling not by increasing the complexity within a single inference, but by increasing the number of search calls. Each call leverages the LLM to make a local, heuristic decision.

To implement this paradigm, we first decompose the global optimization process into atomic subtasks that an LLM can perform more reliably: (1) *solution verification*, which checks the validity of a candidate solution and provides verbal feedback, and (2) *constrained search*, which proposes improvements while respecting problem constraints. These atomic steps are then orchestrated by high-level search frameworks that provide the overall structure: linear reflection loop (Shinn et al., 2023a; Madaan et al., 2023) and Monte Carlo Tree Search (MCTS) (Coulom, 2006).

We evaluate this paradigm on a constructed benchmark of three representative NP problems—Hamiltonian Cycle, Bin Packing, and Traveling Salesperson Problem with Time Windows—a

diverse task variety that ranges from connectivity to spatial and temporal reasoning, each with controllable instance complexity. Our experiments demonstrate that LLM-guided search maintains polynomial-time scalability and continues to deliver competitive approximate solutions as instance size increases. These findings establish a viable path for leveraging P-time computation for intractable problems. In this paper, our main focus is to *explain and quantify* how test-time scaling, achieved by structuring and increasing the number of inference calls, can extend the range of NP instances that P-time machines can effectively address.

In a nutshell, our major contributions are:

1. A new framing that positions LLMs as P-time heuristic components within a structured search process, which achieves scalability by increasing the number of inference calls rather than the complexity of a single one.
2. A decomposition of the overall optimization process into the core subtasks of *verification + constrained search*, enabling fine-grained analysis of how an LLM's capabilities can be effectively orchestrated.
3. Empirical evidence demonstrating that this approach achieves robust polynomial scalability, delivering strong approximate solutions on complex instances where direct prompting fails.

## 2 RELATED WORK

**Language models and NP-hard problems.** Recent efforts to apply LLMs to NP problems can be categorized into three main approaches. The first, direct solving, prompts models to produce a full solution in one pass using techniques such as chain-of-thought or in-context learning. However, studies such as CP-LLMs-ICL (Michailidis et al., 2024) suggest that direct solving is brittle and does not scale with the complexity of the problem. The second approach treats LLMs as *translators*, converting natural language problem descriptions into formal modeling (e.g., MiniZinc (Nethercote et al., 2007), XCSP3 (Boussemart et al., 2016) and CPMpy (Guns, 2019)) for dedicated solvers. While effective, this offloads the core reasoning to external tools, using LLMs only for parsing (Song & Cohen, 2025). Similarly, code-generation pipelines, e.g, the one evaluated with the EHOP dataset (Duchnowski et al., 2025), use LLMs to write solver code, again offloading the actual optimization to code execution. However, as the size and complexity of the NP instance grow, the downstream solver itself can require prohibitive runtimes even when the translation is correct (which is not always guaranteed). The third category involves learning-based methods, where models are fine-tuned on NP instances. However, evaluation results on benchmarks like NPHARDEVAL (Fan et al., 2024a) and its multimodal extension NPHARDEVAL4V (Fan et al., 2024b), have shown that these models often struggle with generalization to instances beyond their training distribution. Balancing the generalization and NP solving capability involves a dilemma in training mixture. Collectively, these works suggest that direct-solving and training-based approaches are not scalable approach to solve NP problems. Our work heads a distinct path. Instead of relying on external solvers or specialized training, we explore test-time scaling by using LLMs as atomic heuristic functions. This aligns with recent agentic reasoning frameworks, e.g, reflection (Shinn et al., 2023a; Madaan et al., 2023) and search dynamics bootstrapping (Lehnert et al., 2024), but focuses on the challenging NP optimizations. By systematically scaling reasoning at test time, we show how to make a P-time machine behave like a NP solver.

**Test-time scaling.** Scaling laws has become one of the most crucial properties of modern large language models. Going beyond the standard model sizes, scaling the length of reasoning traces at test time shows effectiveness in improving model performance (Li et al., 2024; Snell et al., 2024; Muennighoff et al., 2025). Chain-of-thought prompting (Wei et al., 2022; Kojima et al., 2022) provides a simple mechanism for eliciting stepwise reasoning, which helps LLMs break down complex problems into intermediate steps, improving both accuracy and interpretability. More recent approaches leverage self-reflection and iterative refinement, where models re-examine their own outputs (Shinn et al., 2023b; Yao et al., 2023). Methods that integrate search with LLM reasoning – such as Monte-Carlo Tree Search (MCTS) or A* guided by model heuristics – have been shown to improve performance on combinatorial tasks by balancing exploration with model-generated evaluation (Meng et al., 2024; Xie et al., 2024).

**Determinism versus nondeterminism.** Chain-of-thought prompting equips transformers with the ability to perform longer serial computations (Li et al., 2024), but each response still follows a single

deterministic trajectory. Errors therefore accumulate across tokens, making long outputs increasingly fragile. In contrast, when multiple reasoning paths are explored in parallel (Yao et al., 2023; Xie et al., 2023; Zhuang et al., 2023), the process is closer to nondeterministic computation: only one correct path needs to succeed. This distinction mirrors the gap between deterministic and nondeterministic machines, and helps explain why single-pass reasoning struggles on harder NP problems, whereas structured exploration greatly improves success rates.

## 3 POSITIONING P-TIME MACHINES ON NP SPECTRUM

NP problems inherently involve proposing a solution, verifying it, and heuristically editing it towards optimality. Each of these steps, depending on concrete tasks, has different runtime complexity. We view LLMs as a P-time circuit to approximate these underlying tasks, albeit they can be NP ones, e.g., verifying a solution can be easy (P-time) but proposing or even editing one can be hard (NP). With a given solution (as in Fig. 1), we prompt LLMs to check the validity of the solution based on a set of task-specific

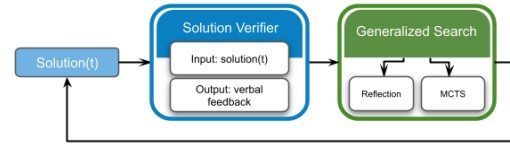

Figure 1: Abstraction of our scaling method. Inside of generalized search module, a single-step search call is performed, e.g., one step of reflection / tree search.

criteria. If deemed invalid, LLMs will generate a verbal feedback[1]. With the solution and its feedback, LLMs perform one step of search. The goal of this search is to refine the input solution based on the verbal feedback. Finally, the refined solution goes through verifier again to continue the loop.

The above test-time searching logic can be seen as an extension of the P-time circuit. This is because both autoregressive decoding and the add-on searching steps remain polynomial. In practice, we consider reflective search and more general MCTS step. The former adds a linear number of LLM reasoning call. The latter at most performs one step of Monte Carlo tree expansion (constant time) and one rollout action (linear time), thus does not break the P-time circuit of LLMs. Thus, the P-time complexity of LLMs is maintained.

### 3.1 VERBAL REASONING BENCHMARK

To test our scaling framework, we construct 3 NP problem sets that are typical and reducible to a broad spectrum of other NP problems. Problems are synthetically generated and solved by open-sourced mixed integer programming (MIP) solvers. Optimal solutions are generally not unique, thus those from the solvers are used as a reference to programmatically evaluate LLM generated ones. Finally, to construct prompt data, we instantiate each instance by a hand-written template that takes the synthetically generated problem parameters. Fig. 2 illustrates an example for each problem set.

**Comparison to prior data.** Comparing to NPHardEval (Fan et al., 2024a) and EHOP (Duchnowski et al., 2025), our task choices introduces extended number of nodes and extra difficulties. Our Hamiltonian cycle scales from 10-50 nodes. Bin packing in this work involves 2-D spatial reasoning instead of 1-D volume. And finally, our TSP examples involve temporal reasoning, which is more complex and realistic. Importantly, we prohibit code generation in the final answer to "enforce" LLMs to reason in the verbal space. However, LLMs are free to use code as intermediate thoughts.

| Task | #node | extra dim | decomposed | verbal-only |
|---|---|---|---|---|
| HAMLT. CYCLE | 10-50 | – | ✓ | ✓ |
| BIN PACKING | 3-20 | 2D spatial | ✓ | ✓ |
| TSP W/ TIME | 7-15 | temporal | ✓ | ✓ |

Table 1: Comparison to NPHardEval benchmark Fan et al. (2024a), we curate a benchmark with higher complexity, extra reasoning dimension, as well as decomposed subtasks (as in Fig. 1).

Below, we summarize the optimization objective for each task:

---

[1]Such as a box being placed multiple times in bin packing, or a flight is taken too early in TSP with Time.

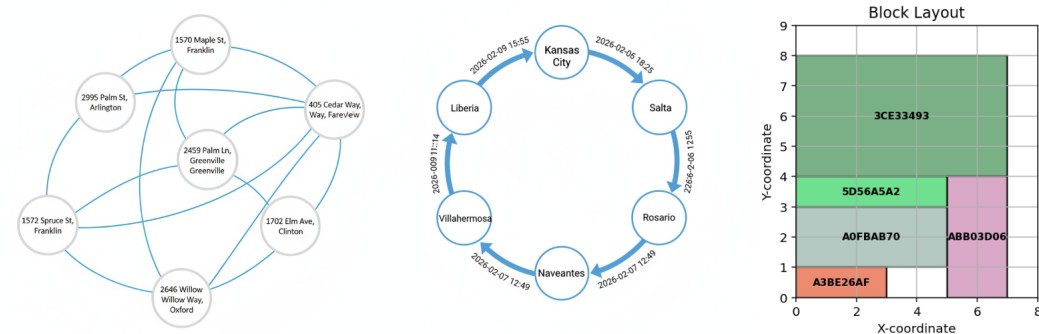

Figure 2: Illustration examples of the three major tasks: *left*: Hamiltonian cycle; ***middle***: TSP with time windows; ***right***: 2D bin packing with cost.

**Hamiltonian cycle.**    With an undirected graph $G = (V, E)$, the goal is to determine a single cycle that visits all vertices exactly once and returns to the starting vertex:

$$x^\star = \arg \min_{x \in \mathcal{C}} \sum_{\{i,j\} \in E} x_{ij}$$

where $x_{ij} \in \{0, 1\}$ denotes whether edge $\{i, j\}$ is included in the cycle; and $\mathcal{C}$ denotes the feasible space enforcing the following constraints: each vertex has exactly two incident selected edges; at least two edges connect any proper subset of vertices to its complement (subtour elimination); and only edges in $E$ can be used, with all $x_{ij}$ being binary.

**2D Bin packing.**    The problems is to fit a set of variable shapes of 2D packages in a bin, each with a value, 2D dimensions, and an option to rotate by 90 degrees. The goal is to maximize the total fit value of packages:

$$x^\star = \arg \max_{x \in \mathcal{C}} \sum_{i,a,b} c_{a,b}^i x_{a,b}^i \tag{1}$$

where $x_{a,b}^i$ denotes whether the $i$-th object (with potential rotation) with bottom-left corner located at position $(a, b)$ when viewing the bin as a 2D quadrant and $c_{a,b}^i$ denotes the package value. The restricted space $\mathcal{C}$ prohibits the overlapping and outbound placements of packages.

**TSP with time.**    As a well-established and more difficult variant of TSP, TSP w/ time injects the temporal cost to each travel edge and a minimum amount of time to spend at each location. The goal is to minimize the total monetary travel cost while meeting the additional time requirement.

$$x^\star = \arg \min_{x \in \mathcal{C}} \sum_{i,j \in \mathcal{L}; k} a_{ij}^j x_{ij}^k \tag{2}$$

where $\mathcal{L}$ denotes the set of locations; $x_{ij}^j$ denotes $\mathbb{1}\{$whether departing from location $i$ to $j$ at time $k\}$; $c_{ij}^k$ denotes the cost; and $\mathcal{C}$ is the edge space that complies TSP connectivity and temporal constraints. There are many ways to formulate the search space $\mathcal{C}$. Irrespective of the concrete formulation, they should cover 1) one-in-one-out for each location; 2) no disconnected sub-tours; and 3) the gap between departure and arrival must accommodate a desired time window.

### 3.2    EVALUATION CRITERIA

The decomposed tasks in Sec. 3 offers a holistic view of LLMs as a P-time machine on NP problem landscape. For *solution verification*, we use an oracle program to check each constraint in the problem-specific constraint set $\mathcal{C}$. If any is violated, the solution is invalid, and valid otherwise. Similarly, for *constrained search*, given an invalid solution and a heuristically generated verbal feedback, we check if the output solution becomes valid. Finally, for the *optimization*, we use the optimal objective value as a reference to calculate a normalized optimization score. For evaluation details, refer to Sec. 4.

**Normalized optimality score (NOP).** Judging whether LLM solutions are optimal yields a sparse binary metric which does not tell how close a LLM is to finding the best answer. To have a smooth evaluation, we can view an optimization as $x^\star = \arg\min_{x \in \mathcal{C}} \sum_i a_i x_i$ and a continuous score can be derived:

| Task | Metric | Solver Help |
|---|---|---|
| VERIFICATION | Validity | ✗ |
| CONSTRAINED SEARCH | Validity | ✗ |
| OPTIMIZATION | NOP | ✓ |

Figure 3: Measurement of 3 tasks in Fig. 1.

$$\mathrm{NOP} = \Big(\frac{a \cdot x'}{a \cdot x^\star}\Big)^s \in [0, 1] \tag{3}$$

where $s = -1$ for $\arg\min$ optimizations and $s = 1$ for $\arg\max$ ones. Note that if a solution violates any problem-specific constraints, we still zero out the NOP score. Without the loss of generality, we use accuracy as $\mathbb{1}(x' = x^\star)$ if the task does not involve an objective value (e.g., Hamiltonian cycle).

# 4 SCALING OF P-TIME LLMS ON NP PROBLEMS

For heuristic-based optimization, scaling is iterative reasoning, where LLMs samples an action, get feedback, refine the action, and retry. This is essentially the decomposition in Fig. 1. The performance of such scaling lies directly in the performance of *solution verification* task and *constrained search* task which are one-step reasoning. Therefore, to have sense where current LLMs are in this scaling landscape, we first evaluate these two individual tasks, then move on to iterative reasoning for the *optimization* task.

Reflective search and MCTS as well-established scaling methods does not break the P-time circuits of LLM. The former is a linear search strategy that generalizes chain-of-thought reasoning. The later generalizes tree-of-thought reasoning, albeit non-deterministic polynomial when fully expanding the tree, is polynomial w.r.t. the number of search calls since both roll-out and node expansion behavior are of polynomial complexity.

**Model and test choices.** We aim to use fast LLMs and apply test-time scaling methods on top of them. At the table, we focus on GPT-5-mini (OpenAI, 2025) and Gemini 2.5 Flash (Gemini, 2025). We should note that each one-step reasoning module in this paper can potentially be replaced by more advanced models and reasoning methods (e.g., with heavy thinking). We leave this direction to future works. Furthermore, to have a clear view on the bottleneck of each decomposed tasks in Fig. 1, we isolate the interaction of them and test them separately. Joint testing of a fully autonomous solving agent is an important future work.

## 4.1 SOLUTION VERIFICATION

In practice, models are prone to make suboptimal or even invalid (i.e., constraint-violating) answers. This often happens when treating LLMs as a black-box zero-shot actor. An autonomous solving agent should know how to self-evaluate its prediction and generate feedback for further iteration. To simulate such cases, we mix the ground truth solutions (from MIP solver) and heuristically perturbed ones (with guaranteed constraint violation), giving us 50-50 mixture for the verification task. To see whether verification accuracy degrades w.r.t. problem complexity, we plot LLMs' performance on the easy half and difficult half separately, judging by the number of nodes in the task search space[2].

To avoid the optimal solutions from leaking into LLM inference, we do not use the optimal solution in the solution verification task. Instead, an oracle program is used to check the validity of a candidate and output verbal feedback.

**Perturbation choices.** We randomly sample $\{1, 3, 5\}$ corrupt actions from *add*, *replace*, and *remove* and apply them on a solver-generated optimal solution repeatedly to the point where some task-specific constraints are violated. Since we mix these corrupted solutions with gold, we have a baseline of 50% accuracy.

---

[2]We should note that the actual difficulty of an example varies a lot depending on the number of nodes, edges, and the formed search landscape. Therefore #node is a sheer surrogate for complexity.

**Results.** Even with one-step reasoning, LLMs are good at solving the verification task, as shown in Fig. 4. This behavior meets the common expectation that some NP problems can be verified at P-time. With more aggressive perturbation, LLMs generally have higher accuracy as solutions are corrupted more heavily thus providing more classification signals. The overall degradation w.r.t. the number of perturbations does not vary significantly. Given the fact that LLMs are fast evolving these days, the gap can be closed by more aggressive reasoning method or further tuning.

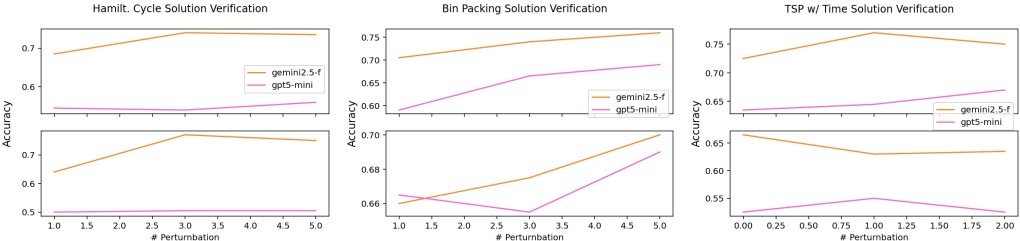

Figure 4: Solution verification of 3 tasks. ***Upper*** row: easy half; ***lower*** row: difficult half.

## 4.2 CONSTRAINED SEARCH

Now, we test whether LLMs can respond to verbal feedbacks to correct invalid solutions. To simulate this scenario, we use an oracle verifier, and at test-time, we allow multiple samples ($k = 4$) to be generated. If LLMs has saturated performance in solution verification, it would easily pick the best-of-4. Thus, if any of the 4 samples is valid, the prediction is deemed accurate.

Our problems involve multiple hard constraints that interact with each other. Therefore, in constrained search, only measuring if the violated constraints are corrected is not enough. Instead, we measure if LLMs can make invalid predictions (with any violation) become valid ones (free of violation, but can be suboptimal).

Similar to how we evaluate the verification task, we simulate this by perturb gold solutions for $\{1, 3, 5\}$ corrupt actions, obtain the verbal feedback, and measure if LLM can generate candidate solutions that fix the identified constraint violations. In practice, proposing a solution has equivalent complexity as fixing one. As in Fig. 5, with more corrupted solutions, constraints are generally more challenging to fix, therefore the corrective rate degrades quickly.

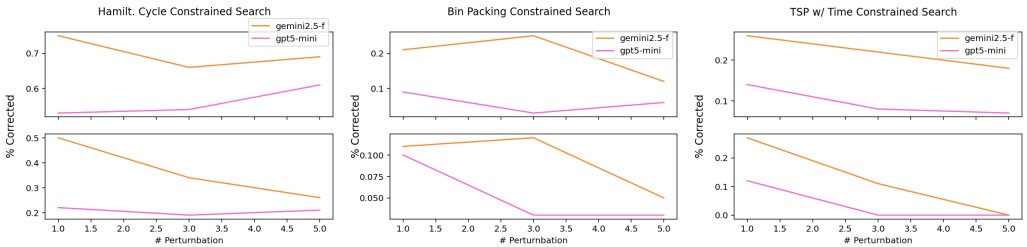

Figure 5: Corrective rate of constrained search. ***Upper*** row: easy half; ***lower*** row: difficult half.

### 4.2.1 GLOBAL OPTIMIZATION

With the single-step tasks in Sec. 4.1&4.2, we move on to iterative optimization. Usually, the ground truth solution is not unique. Therefore, the output should 1) satisfy all hard constraints of the given problem; and 2) have equivalent objective as the reference solution.

To avoid error accumulation, we use the oracle solution verification instead of LLMs. This allows a reliable verbal feedback during iterative search and avoids error accumulation especially considering the headroom in Fig 4 and Fig. 5.

### 4.2.2 REFLECTIVE SEARCH

The first scaling method is reflection which is a linear MDP in Algo. 1. We explore how the model iteratively improves its answer via chained reflections. Feedbacks are directly from the oracle solution verifier, and at each search call, LLMs are asked to improve the incoming solution by either fixing violated constraints or improving objective. The final output is the best solution the search loop ever encountered.

---
**Algorithm 1** Reflective Search
---
1: **Input:** Problem Description $\mathcal{P}$
2: **Initialize:** Solution candidate $x$; Trajectory $\tau$.
3: $x \leftarrow$ LLM.search($\mathcal{P}$)                    ▷ initial state
4: **for** $i = 1$ to #trials **do**
5:     *# Reflection search call*
6:         $r \leftarrow$ LLM.verify($\mathcal{P}, x$)                    ▷ review
7:         $x \leftarrow$ LLM.search($\mathcal{P}, x, r$)                    ▷ refine
8: **end for**
9: **return** $\tau.best$
---

In practice, we let `LLM.search` to generate $k = 4$ action candidates, and greedily sample the best one as the output $x$. Furthermore, we only consider the reflection $r$ in the immediate previous state to have a manageable input length[3]

**Results.** In Fig. 6, we scale LLMs over 20 search steps. The result shows strong performance improvement even for such a simple MDP. Across the 3 tasks and for both GPT=5-mini and Gemini2.5 Flash models, scaling consistently improve optimization by a large margin. Complexity indeed plays a critical role in scaling as the easy half problems have substantially higher performance then the difficult half. Scaling becomes ineffective when the probably most difficult TSP w/ time problems reached 10-15 nodes.

### 4.2.3 MCTS

Comparing to reflective search, we allow the linear reasoning trajectory to grow horizontally depending on the exploration factors. MCTS generalized reflective search since each time MCTS decides to expand on a node, we essentially perform a one step of reflective search. To balance exploration and exploitation, we use the standard UCT score. Individual MCTS search block (mentioned in Fig. 1) is in line 5-11.

**Reward scoring.** To compute UCT score, we need to first calculate the reward of a state $x$. For Hamiltonian cycle, we set reward to 100 if the solution candidate is correct and 0 otherwise. For bin packing, we set the reward to the objective of the state since this is an $\arg\max$ optimization. For TSP with time, since it is a $\arg\min$ problem, we use a constant budget $B$ for all examples and set reward to $B - o(x)$. We should note that the choice of reward function directly impacts the action distribution and thus the tree search results. Our choices on these hyper-parameters yield consistent numbers in our preliminary study.

---
**Algorithm 2** Monte-Carlo Tree Search
---
1: **Input:** Problem Description $\mathcal{P}$
2: **Initialize:** Solution candidates $x$; Trajectory $\tau$.
3: $\tau$.root $\leftarrow$ LLM.search($\mathcal{P}$)                    ▷ initial state
4: **for** $i = 1$ to #trials **do**
5:     *# MC search call*
6:         leaf $\leftarrow$ greedy_sample($\tau$)
7:     **if** is_rolled_out(leaf) **then**
8:             $r \leftarrow$ LLM.verify($\mathcal{P}, x$)                    ▷ review
9:             $x \leftarrow$ LLM.search($\mathcal{P}, x, r$)                    ▷ refine
10:            leaf.expand($x$)
11:     **end if**
12:     LLM.roll_out(leaf)                    ▷ UCT propagation
13: **end for**
14: **return** $\tau.best$
---

**Answer filtering.** We note that LLMs tend to generate a lot of invalid candidate solutions. Sometimes, this causes the tree to be saturated of invalid solutions at early stages of searching, and making the search less efficient since the action distribution scores (UCT) are noninformative. In practice, we found filtering out invalid candidates during the expansion operation consistently outperforms

---

[3]Prior works (Shinn et al., 2023a; Madaan et al., 2023) have shown that using reflection trajectory could yield better results.

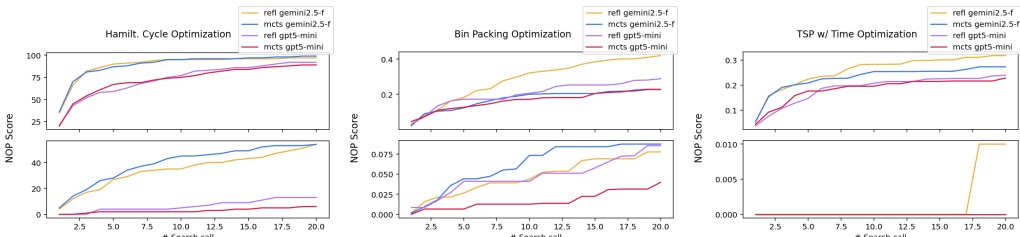

Figure 6: Test-time scaling of reflective search and MCTS. *Upper* row: easy half; *lower* row: difficult half.

retaining all candidates. Again, we use an oracle to do so. In a fully autonomous setting, LLMs can play the same role.

**Tree sizes.** MCTS search configuration involves how many number of action candidates (the same $k$ as in Sec. 4.2.2) and the max depth $d$ of the tree. Aggressive choices on these numbers lead to marginal returns, quickly increasing latency, and context length explosion. From preliminary trials, we found a good choice for trials×width×depth is $20 \times 4 \times 10$. During the roll-out operation, the maximum probed depth is the same as the maximum depth of the tree.

**Results.** With the above configurations, the test-time scaling of MCTS is reported side-by-side with reflective search in Fig. 6. Despite of the general gap between Gemini 2.5 Flash and GPT 5 mini, to our surprise, reflective search works effectively when the problems are on the easy half. MCTS performs better on the difficult half. We hypothesize this is because when problems are easy, linear MDP already provides good signal from previous state while MCTS does not always expand node at every search call, thus falling behind. And when problems are difficult, the exploration behavior of MCTS starts to outperform.

## 5 ANALYSIS

### 5.1 IMPACT OF INITIALIZATION

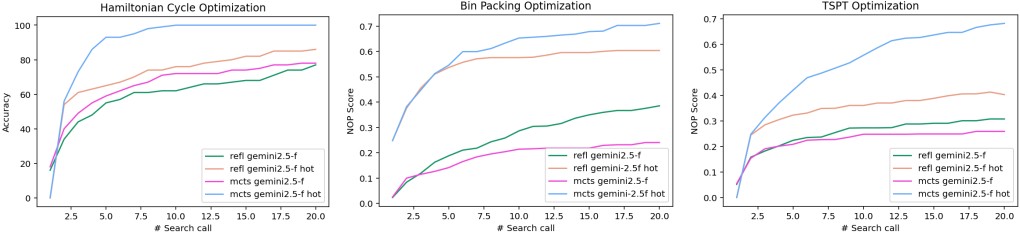

Figure 7: Impact of initialization on Gemini 2.5 Flash on the full dataset (both *easy* and *difficult* halves). *hot* start means the search start from a perturbed optimal solution.

Solution initialization has been a critical technique in the domain of global optimization methods. Our scaling methods is no exception to this. Here, we take Gemini 2.5 Flash and do an ablation on the impact of initial solution by simulating a starting point which is corrupted from the optimal solution given by the solver. Specifically, we use the corrupted solutions with #perturbation=1 in Sec. 4.1 and refer to this test as *hot* start in Fig. 7. To our expectation, a solution close to the optimal one helps by a large margin. The behavior of reflective search and MCTS also differs with hot start. Reflection starts to suffer from limited exploration and MCTS constantly wins by a large margin. For the HC optimization, MCTS even reaches ∼100% accuracy after 10 searches.

## 5.2 SCALING W.R.T. COMPLEXITY

Fig. 8 shows the performance by bucketed problem complexity. At finer-grained look, e.g., on the Hamiltonian cycle, it is not necessary that more complex instances receive worse performance. LLM performances are relatively retained. This can be explained that the Hamiltonian cycle is the easiest of the three problems and our scaling are still effective. For the bin packing and TSP w/ time tasks, there is a clear degradation trend over complexity. However, the drop of performance is rather not flat. As noted in Sec. 4, the number of node/edge is only a weak surrogate of complexity. The actual difficulty of an example depends on the search landscape, explaining the bumpy performance decrease.

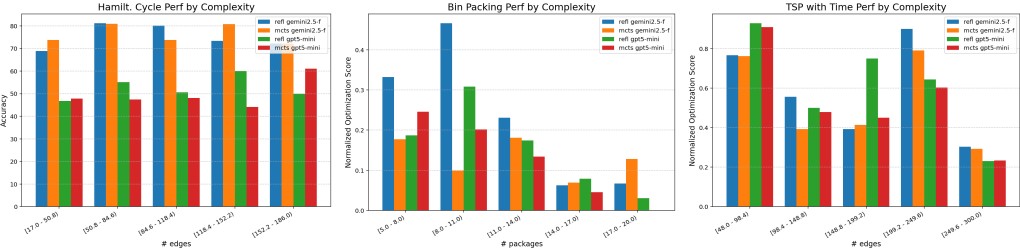

Figure 8: Performance/Complexity Analysis

## 6 CONCLUSIONS

The emergent ability of Large Language Models to tackle NP-hard problems is fundamentally constrained by their P-time nature. In this paper, we confront this challenge by proposing a paradigm shift: instead of deploying LLMs as monolithic solvers or simple translators, we frame them as P-time heuristic components within scalable, classical search frameworks.

Our approach achieves scalability by increasing the number of heuristic search calls, rather than the computational depth of a single inference. By decomposing the problem-solving process into core subtasks of solution verification and constrained search, and orchestrating these with algorithms like Reflective Search and MCTS, we demonstrated robust performance on a challenging benchmark of NP problems. Our empirical results show that this LLM-guided search maintains polynomial-time scalability, consistently delivering strong approximate solutions for instances where direct prompting methods fail.

Furthermore, our analysis revealed the critical role of initialization, where high-quality starting points significantly boost performance, underscoring the powerful synergy between LLM-driven heuristic guidance and structured exploration. These findings establish a viable and principled path for applying P-time computational models to intractable NP landscapes. Ultimately, this work suggests that the most effective role for LLMs in complex reasoning is not as standalone super-reasoners, but as powerful, callable heuristic guides integrated within the robust, scalable architectures of classical search algorithms.

## REPRODUCIBILITY STATEMENT

We have taken multiple steps to facilitate reproducibility of our results. All problem definitions, experimental settings, and evaluation metrics are described in Sections 3–5. We will release a repository containing the complete source code, data preprocessing scripts, and instructions for running all experiments. These resources enable independent researchers to reproduce the empirical comparisons between different heuristic search paradigms and to verify the phenomena reported in the paper.

## ETHICS STATEMENT

Our study involves only synthetic problem instances; no human subjects, private data, or personally identifiable information are used. All algorithms are evaluated in a research context and are not

deployed in safety-critical applications. We will release code and generated instances solely for reproducible scientific research and under an open-source license that respects data privacy and legal compliance. We are unaware of any foreseeable negative societal impacts beyond those generally associated with advances in automated reasoning and large language models, and we have taken care to document experimental settings and limitations to support responsible follow-up work.

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

## A    USAGE OF LARGE LANGUAGE MODELS (LLMS)

During the preparation of this work, the authors utilized Large Language Models (LLMs) to support several non-scientific aspects of the research and writing process. The specific applications are as follows:

**Code Generation**: LLMs (e.g., OpenAI ChatGPT) were used to draft small helper scripts for data visualization and figure rendering (e.g., plots shown in Fig. 2). All generated code was manually reviewed, debugged, and integrated by the authors.

**Writing Assistance**: LLMs were occasionally consulted to propose alternative phrasings for abstracts, section headings, and figure captions. These suggestions were treated as editorial input and were carefully reviewed and revised by the authors before inclusion. No LLM was used to generate research ideas, experimental results, proofs, or other scientific contributions. All conceptual advances, dataset construction, algorithm design, and analysis were performed solely by the authors.

## B    PROMPT DESIGN AND IMPLEMENTATION DETAILS

This appendix details the prompt engineering, feedback generation, and LLM-based search algorithms (MCTS and Reflection Search) used in our experiments on three NP-hard problems: Hamiltonian Cycle (HC), Bin Packing (BP), and Traveling Salesman with Time (TSPT). All prompts were automatically constructed from structured instance data and fed to the LLM without human post-editing.

### B.1    HAMILTONIAN CYCLE (HC)

**Task.**    Given an undirected graph of logistics hubs, the goal is to design a route that starts at any hub, visits every hub exactly once, and returns to the starting hub.

> You are a helpful planning assistant to design a routing path that connects logistics hubs. The goal is to have a route that starts at any given hub, visits each hub exactly once, and returns to the same starting hub. Below is a hub connection graph ...

**Base Prompt.**    The LLM outputs a JSON list of hub names matching the exact strings in the problem.

**Feedback.**    Candidate routes are evaluated by a deterministic checker that returns status codes such as `MISMATCH`, `DISCONNECTED`, `DUPLICATE`, `TOO_MANY`, `TOO_FEW`, or `OK`. These codes are verbalized into natural-language feedback, for example:

> – Your route can not reach hub B since it is not directly connected to hub A.

**Monte Carlo Tree Search (MCTS).**    Each MCTS node corresponds to a candidate cycle. At expansion, the feedback-augmented prompt requests $k$ improved solutions:

> [ ["hub1", "hub2", ...], ... ]

Candidates are re-evaluated and assigned a reward of 100 if valid (`OK`) or 0 otherwise. Typical parameters: `num_trials=20`, `max_width=4`, `max_depth=10`.

**Reflection Search.**    Instead of exploring a tree, Reflection Search performs single-path refinement:

> Given the above feedback(s), suggest k different and improved solutions. Output each of your solution as a separate list of hub names and put them in the following json format: [ candidate solution 1, candidate solution 2, ... ]

The environment maintains only the current best route. At each step the LLM proposes $k$ new candidates, each is evaluated and the best is chosen greedily until the maximum number of trials is reached.

## B.2    BIN PACKING (BP)

**Task.**    Pack a set of rectangular packages into a fixed-size 2-D bin to maximize total value without overlap. Each package may be rotated.

> Arrange the following packages inside a rectangular bin of size [WIDTH]x[HEIGHT] to maximize the total value. Each package may be rotated. Output your arrangement as a JSON object where each key is a package ID and the value specifies the (x,y) position and whether the package is rotated.

**Base Prompt.**

**Feedback.**    The evaluator checks boundary (`OOB_id`), overlap (`OVERLAP_id`), existence (`NONEXISTING_id`), and value optimality (`TOO_FEW`, `TOO_MANY`, `SUBOPTIMAL`, `OK`), and produces natural-language feedback, e.g.,

> – Package P3 overlaps with other packages.
> – The arranged solution has too few packages. Try to fit more.

**Monte Carlo Tree Search (MCTS).**    At each node the LLM is queried for $k$ improved arrangements:

Rewards are given by the total value if all constraints are satisfied.    Typical parameters: `num_trials=20`, `max_width=4`, `max_depth=10`.

**Reflection Search.**    Reflection Search refines a single candidate arrangement:

> Upon closer checking on the solution, you found some potential issues with it. Specifically, <feedback>

> Given the above feedback(s), suggest k different and improved solutions. Output each of your solution as a separate element in a list and put them in the following json block: [ candidate solution 1, candidate solution 2, ... ] where each candidate is a solution in its own format.

At each iteration the best arrangement (highest value, valid constraints) is chosen and passed back for further improvement.

## B.3    TRAVELING SALESMAN WITH TIME (TSPT)

**Task.**    Plan a minimum-cost round trip that starts and ends at the same city, visits each other city exactly once, and satisfies minimum stay-time and budget constraints.

> You are a helpful trip planning assistant to help a user plan a trip that minimizes the overall cost subject to certain constraints. The user wants to take a tour of {LOCATIONS} from {START_DATE} to {END_DATE} ({N} days). The trip starts and ends at {ROOT_CITY}. Flight options are given as "DEPARTURE/ARRIVAL/DEPARTURE_TIME":  "cost":..., "duration": HOUR:MINUTE .

**Base Prompt.**

**Feedback.**    The evaluator checks TSP constraints (one-in/one-out, no subtours, no back–forth trip), time constraints (minimum stay hours), and cost constraint. Violations are verbalized, e.g.,

> – The planned solution does not meet the minimum stay time requirement for Philadelphia.
> – The planned solution exceeds the budget of 5000.

**Monte Carlo Tree Search (MCTS).**    At each node the LLM is prompted for $k$ improved itineraries:

[ [flight1, flight2, ...], ... ]

Rewards are defined as budget − cost if all constraints are satisfied, otherwise 0. Typical parameters: `num_trials=20`, `max_width=4`, `max_depth=10`.

**Reflection Search.** Reflection Search sequentially refines a single candidate trip:

Upon closer checking on the solution, you found some potential issues with it. Specifically, <feedback>

Given the above feedback(s), suggest k different and improved solutions. Output each of your solution as a separate list of flight keys and put them in the following json format: [ candidate solution 1, candidate solution 2, ... ]

At each step the LLM proposes $k$ new itineraries, each is evaluated for cost and constraints, and the best candidate is selected for the next round.

B.4    IMPLEMENTATION NOTES (ALL TASKS)

- **Model:** OpenAI `gpt-5-mini`, 128k context, reasoning mode set to `minimal`.
- **Parsing:** Candidate solutions are extracted from the final `json` block using a regex-based parser with normalization (single quotes and boolean values converted to valid JSON).
- **Logging:** All prompts, LLM responses, evaluator feedback, and search trajectories are saved for reproducibility.

