# OpenReview forum: "Polynomial-Time Reasoning at the Edge of NP"
_ICLR.cc/2026/Conference — ICLR 2026 Conference Withdrawn Submission_

### Official Review · Reviewer_Z8Hq · 2025-10-28

**Soundness:** 2
**Presentation:** 2
**Contribution:** 1
**Rating:** 2
**Confidence:** 3

**Summary:**

This paper applies existing LLM-based search methods (reflection and MCTS) to NP-hard problems, framing them through a complexity-theoretic lens. The authors argue that LLMs are "P-time machines" and propose using them as heuristic components within classical search algorithms rather than as monolithic solvers. They decompose the optimization process into verification and constrained search subtasks, then orchestrate these with reflection loops and MCTS.

Experiments on three synthetic NP problems (Hamiltonian Cycle, 2D Bin Packing, TSP with Time Windows) show improvement over direct prompting baselines. The authors claim this demonstrates "polynomial scalability," though performance degrades significantly as problem size increases. A key finding is that initialization quality strongly impacts performance.

**Strengths:**

- **Clear exposition**: The writing is generally clear and the P-time framing, while not deeply novel, provides a clean way to think about LLM capabilities and limitations.

- **Systematic evaluation structure**: Breaking down evaluation into verification, constrained search, and full optimization provides useful diagnostics about where LLMs succeed and fail.

- **Initialization analysis**: The hot start experiments (Fig 7) provide useful insights about the importance of solution quality and the interaction between MCTS and reflection in different regimes.

- **Honest about limitations**: The authors acknowledge using oracle verification to avoid error accumulation and note that node count is only a "weak surrogate" for complexity.

**Weaknesses:**

- **Limited novelty**: The core contribution appears to be reframing existing search techniques through P vs NP terminology rather than introducing fundamentally new methods. The paper essentially applies well-established LLM search techniques (reflection, MCTS) to NP problems and discusses the results in terms of complexity theory.

- **Theoretical contribution is thin**: The P-time complexity observation is straightforward, and the connection to actual P vs NP theory remains fairly surface-level without engaging with approximation complexity or deeper computational questions. Contribution 1 ("new framing") amounts to relabeling LLMs "P-time heuristic components" and observing that increasing search calls maintains polynomial complexity - neither insight feels particularly novel. Contribution 2 (decomposing into verification + search) has been explored extensively in prior work (e.g., Tree-of-Thoughts, Reflexion, etc.). Beyond the basic framing, there's little in the way of formal analysis (e.g.,  approximation guarantees, sample complexity bounds, or characterization of failure modes) that would deepen the theoretical contribution.

- **Oracle verification limits practical relevance**: The optimization experiments (Sec 4.2.1) use oracle verifiers for "reliable verbal feedback." But Fig 4 shows LLM verification achieves only 60-80% accuracy. It's unclear how these errors would compound over 20 iterations in a fully autonomous system. The gap between the oracle-assisted results and what's actually achievable seems important but isn't explored.

- **Scalability claims feel overstated**: Performance degrades notably on harder instances. For TSP with time at 10-15 nodes (still quite small), NOP drops to near-floor-level (Fig 6). The method doesn't demonstrate success on problem sizes where you'd typically need sophisticated heuristics. The "polynomial scalability" framing is technically correct but somewhat misleading given the practical limitations.

- **Missing key baselines**: No comparison to classical optimization heuristics (greedy + local search, simulated annealing, etc.) or to learning-based methods. With only direct prompting as a baseline, it's hard to assess whether LLMs provide value over simpler alternatives.

- **Experimental rigor could be stronger**: Only 2 models tested, no error bars or significance tests, synthetic benchmark without validation on standard OR suites, crude easy/difficult split by node count, no runtime analysis despite "polynomial time" claims.

- **Benchmark scope**: Problems are synthetically generated and solved by MIP solvers. It's unclear whether these instances are representative of NP-hard search problems in real-world settings.

- **Presentation details**: Several typos and formatting issues (e.g., "Perturnbation" in Fig 4, misplaced legends) suggest the paper could use another editing pass.

**Questions:**

1. Regarding lines 260-262: You mention avoiding "optimal solutions from leaking into LLM inference" as justification for using an oracle verifier instead of the optimal solution during verification. However, if each solution (whether ground truth or perturbed) is evaluated independently by the LLM, it's unclear how information could "leak" between examples. Could you clarify what specific leaking concern you're addressing here? Is this about the prompt design, the evaluation protocol, or something else?

2. Could you provide results using LLM-based verification for the full optimization pipeline? It would be helpful to understand the performance gap between oracle and LLM verification to assess real-world viability.

3. How does your approach compare to standard classical heuristics (e.g., greedy construction + local search)? This would help clarify whether LLMs provide meaningful value over simpler baselines.

4. Your more difficult instances are still relatively small by optimization standards (e.g., TSP with 10-15 nodes). Can you characterize more precisely where the method breaks down - what's the largest instance size you can solve to reasonable quality?

5. The initialization experiments show strong effects from hot starts. Can you provide more detail on how initialization quality affects performance, perhaps with a range of perturbation levels or random initialization?

6. Could you include some runtime analysis to complement the complexity claims? It would be useful to see how wall-clock time scales with problem size compared to classical methods.

---

### Official Review · Reviewer_zkVn · 2025-11-01

**Soundness:** 3
**Presentation:** 3
**Contribution:** 2
**Rating:** 2
**Confidence:** 4

**Summary:**

The paper explores how large language models (LLMs), though limited to polynomial-time computation per call, can still tackle NP-hard problems through structured test-time reasoning. Prior approaches such as CoT and ToT use LLMs as heuristic searchers but lack a clear computational framing or principled decomposition. The authors propose a polynomial-time reasoning framework that treats the LLM as a bounded-time heuristic embedded in a structured search loop with two components: (1) an oracle-based verifier that checks and critiques candidate solutions, and (2) a constrained search module using reflective refinement or MCTS. This setup preserves polynomial complexity per call while scaling reasoning through repeated guided trials. Experiments on Hamiltonian Cycle, 2D Bin Packing, and TSP show that the approach improves over single-shot prompting, with MCTS and better initialization yielding the strongest gains.

**Strengths:**

* The paper is clearly written and well organized.
* The idea of combining LLMs with structured search is presented clearly and implemented carefully.
* The experimental setup is detailed and reproducible.
* Results show that scaling the number of reasoning calls improves performance.

**Weaknesses:**

1. The novelty is limited. The proposed setup (LLM + external verifier + structured search) has already appeared in many prior works such as [1, 2]. These studies also combine an LLM heuristic, structured search, and an oracle or executable verifier, which is nearly identical to this paper’s design. The only new element here is the “polynomial-time reasoning” framing, which is conceptual rather than technical.

* [1] Zheng et al., Monte Carlo Tree Search for Comprehensive Exploration in LLM-Based Automatic Heuristic Design, ICML 2025.
* [2] Zhao, Lee, and Hsu, Large Language Models as Commonsense Knowledge for Large-Scale Task Planning, NeurIPS 2023.

---

2. The “polynomial-time” claim is mainly conceptual, without any formal analysis of computational complexity, convergence, or scaling behavior.

---

3. There are no direct comparisons against existing Tree-of-Thought, Reflexion, or LLM-MCTS methods on shared datasets, nor against classical solvers such as OR-Tools or CP-SAT.

---

4. The benchmarks (Hamiltonian Cycle, Bin Packing, TSP) are self-constructed and synthetic. Without standard baselines, it is difficult to assess their difficulty or interpret the reported performance. In addition, the “verbal-only” output format further limits relevance to real-world reasoning or tool-use scenarios.
Figure 2 is not really informative. It seems to show standard NP problems but does not explain them intuitively, and the IDs and numeric labels in the figure are not informative.

---

5. Figures 4 and 5 have cluttered horizontal axes. Only the integer values (1, 3, 5) are meaningful, while the many floating-point tick marks in between add visual noise and make the plots harder to read.

---

6. The writing of the method and experiments is somewhat mixed, which makes the overall structure harder to follow. Section 3 Polynomial-Time Reasoning Framework mixes the conceptual motivation, in the beginning of the section, with dataset and task descriptions, so it is not always clear where the method explanation ends and the experiment setup begins. Likewise, Section 4 presents results while also introducing key procedural details, such as search configurations and verifier behaviors, that might be better placed earlier when the framework is first introduced. Separating the algorithmic description (Sections 3–4) from the experimental evaluation (Section 5) would make the presentation clearer and easier to understand.

**Questions:**

1. How does this approach fundamentally differ from prior frameworks, which also combine LLM reasoning, external verification, and structured search?

2. Could you clarify again why those standard NP problem benchmarks (e.g., NPHardEval, EHOP) are not used for evaluation? I am not sure if I captured the correct reason in the paper.

3. Is it possible to make quantitative comparisons with classical solvers (e.g., OR-Tools, CP-SAT) or other LLM-based approaches to show the practical gap?

---

### Official Review · Reviewer_hob8 · 2025-11-01

**Soundness:** 2
**Presentation:** 2
**Contribution:** 2
**Rating:** 2
**Confidence:** 4

**Summary:**

This paper studies the performance of Large Language Models (LLMs) in solving NP problems. The authors consider three tasks: solution verification, constrained search for correcting infeasible solutions and scaling for global optimization. On Hamiltonian cycle, TSP with time
windows and 2D bin packing with cost problems, the authors study the scaling behavior and the importance of initial solutions. Particularly, compared to directly prompting, step-wise search (either reflective search or MCTS) delivers better performance.

**Strengths:**

- The paper comprehensively evaluates the performance of LLMs in solving NP problems, spanning from solution validation, correction to optimization
- The paper is generally well written

**Weaknesses:**

- I think the paper does not position itself well. Generally, it does not propose new algorithms or train new models, which makes it more like a benchmark paper. But it also doesn't like common benchmark & dataset papers that publishes carefully curated artifacts. Perhapse the author could reconsider the motivation of the paper.
- The tasks it studies are largely overlapped with [1], reducing the novelty of the paper.
- The findings of the research are trivial or known. Particulaly, "good initial solution usually lead to better final results" is widely observed in optimization community; Scaling improves vanilla CoT is also widely acknowledged.
- The details of NP problems generation are not given in main text. The criteria of categorization "easy" and "hard" are unclear.
- The scenario of using LLMs to solve NP problems is less realistic, especially considering the well-established optimizers (e.g., Gruobi) that return very good solution in seconds to minutes. Also, the context window limit makes LLMs unsuitable for industry-level problems.

[1] PlanBench: An Extensible Benchmark for Evaluating Large Language Models on Planning and Reasoning about Change

**Questions:**

see Weakness

---

### Official Review · Reviewer_F2WK · 2025-11-07

**Soundness:** 2
**Presentation:** 1
**Contribution:** 1
**Rating:** 2
**Confidence:** 3

**Summary:**

This paper studies the ability of LLM reasoning to tackle NP problems. It uses a framework where the LLM generates an answer as a step which is then verified and iterated upon. The iteration search policy could be self-reflection or MCTS. The authors argue that this setup does not break the P-time complexity but succeeds at some NP tasks.  There is a severe lack of clarity.

**Strengths:**

* The reasoning tasks chosen are decently complex
* The explanation and analysis in the experiments section is detailed

**Weaknesses:**

* The writing is unclear and I am confused about some fundamental aspects:
  * What is the contribution here? Chain of thought and tree of thought are already multi-step reasoning methods. Self-reflection, MCTS and search/verify architectures are not new in this context. You mentioned that each step in your method could correspond to a full chain of thought but this was not implemented in the paper. And even if it was, what would be the point of that? I struggle to see the difference between this method and any chain of thought style method so this part is not explained well.
  * There is no methodology section or preliminaries / formalisation of the problem, all of which might have helped me understand the problem setting and motivation.
* The use of LLMs for combinatorial problems like TSP is not well motivated in the paper. It would have been interesting to see difficult reasoning tasks that were based in dense natural language, as this is where LLMs are useful.
* It’s hard to know exactly what is being evaluated in the experiments. What are the questions you’re trying to answer?
* Allowing coding during reasoning time defeats the whole purpose of LLM reasoning, especially since there are many code solutions to these combinatorial problems available in the pre-training data.
* The method is not compared against any baselines.
* The whole paper is riddled with grammar and spelling mistakes, which are distracting and make it sound unprofessional. Please use an automated tool to fix these.

**Questions:**

See weakness.

---

### Note · Authors · 2025-11-13

I have read and agree with the venue's withdrawal policy on behalf of myself and my co-authors.